# eHomeSeniors Dataset: An Infrared Thermal Sensor Dataset for Automatic Fall Detection Research

**DOI:** 10.3390/s19204565

**Published:** 2019-10-21

**Authors:** Fabián Riquelme, Cristina Espinoza, Tomás Rodenas, Jean-Gabriel Minonzio, Carla Taramasco

**Affiliations:** 1Escuela de Ingeniería Civil Informática, Universidad de Valparaíso, Valparaíso 2340000, Chile; tomas.rodenas@uv.cl (T.R.); jean-gabriel.minonzio@uv.cl (J.-G.M.); 2Centro de Investigación y Desarrollo en Ingeniería en Salud, Universidad de Valparaíso, Valparaíso 2340000, Chile; 3Independent Researcher, Valparaíso 2340000, Chile; cristina.espinozalai@gmail.com

**Keywords:** fall detection, public dataset, thermal sensor, infrared sensor, smart home

## Abstract

Automatic fall detection is a very active research area, which has grown explosively since the 2010s, especially focused on elderly care. Rapid detection of falls favors early awareness from the injured person, reducing a series of negative consequences in the health of the elderly. Currently, there are several fall detection systems (FDSs), mostly based on predictive and machine-learning approaches. These algorithms are based on different data sources, such as wearable devices, ambient-based sensors, or vision/camera-based approaches. While wearable devices like inertial measurement units (IMUs) and smartphones entail a dependence on their use, most image-based devices like Kinect sensors generate video recordings, which may affect the privacy of the user. Regardless of the device used, most of these FDSs have been tested only in controlled laboratory environments, and there are still no mass commercial FDS. The latter is partly due to the impossibility of counting, for ethical reasons, with datasets generated by falls of real older adults. All public datasets generated in laboratory are performed by young people, without considering the differences in acceleration and falling features of older adults. Given the above, this article presents the eHomeSeniors dataset, a new public dataset which is innovative in at least three aspects: first, it collects data from two different privacy-friendly infrared thermal sensors; second, it is constructed by two types of volunteers: normal young people (as usual) and performing artists, with the latter group assisted by a physiotherapist to emulate the real fall conditions of older adults; and third, the types of falls selected are the result of a thorough literature review.

## 1. Introduction

The continually aging population worldwide [1] represents a huge challenge for the care and prevention systems of accidents within the home, especially for the elderly living alone. A permanent risk in older people are falls [2] (In specialized literature, it is indicated that, on average, about one third of adults over 65 suffer a fall a year. Actually, although we know that falls are very frequent in older adults, after looking for the origin of this sentence, we have not been able to arrive at a concrete and updated reference where this is proven). The risk of falls and their negative effects on health increase with age. A study of 110 adults older than 90 years showed that only one half who fall are capable of getting up on their own [3]. Falls can lead to various health problems in the short and long terms, such as fractures [4], carpet burns, dehydration, hypothermia, pneumonia [3], volume depletion, internal infections and bleeding, cellulitis, ulcers, chest pain, syncope, heart attacks, and even death [5]. From a psychological point of view, many elderly people after falling develop a fear of falling again, which limits their daily activities [6].

Due to the above, during the last ten years, various fall-detection systems (FDSs) have been developed, both for the detection and early assistance of falls for the elderly and for the prevention and prediction of falls in their activities of daily living (ADL) [7]. FDSs are computational algorithms usually based on either a predictive or a machine-learning approach. Therefore, they require a training dataset, which allows them to differentiate a real fall from normal activities out of risk, such as walking, standing, sitting, etc.

The main ways to collect fall datasets are wearable devices and ambient-based sensors. Table 1 illustrates the main positive and negative aspects of each type of device [8,9]. Among the different ambient-based sensors, infrared thermal sensors allow to capture data even during no-light conditions. Moreover, some studies have concluded that it is easier to analyze thermal rather than normal images [8]. For data analysis, the collection of quality data is often a costly problem. In the context of elderly falls, there are additional ethical issues, the most critical of which is that one cannot ask an old person to fall voluntarily due to the high risk of injury. As we shall see in Section 2, since 2008, some public datasets on falls have being published to use as benchmarking and training of new FDS. This has undoubtedly been a great help for research in the area. However, fall datasets still present some general deficiencies:Fall datasets are still few, as we will see in Section 2.Due to the ethical problems mentioned above, the datasets do not include elderly falls but falls of healthy young volunteers, who fall differently compared to older adults. The most noticeable difference is that young people fall with a greater acceleration than the elderly [9]. Other kinesthetic differences will be described in more detail in Section 3.2. Because of this, the performance of many algorithms could drastically decrease by changing their laboratory environment to that of a real environment (i.e., the elderly home).Many fall datasets are based on acceleration data, which has been shown to be insufficient on its own as predictors of falls. In fact, it has been proven that FDSs based on acceleration amplitude produce a large number of false alerts unless post-fall posture identification is also considered [10].Although datasets based on video recordings often use low-resolution images (e.g., depth images with 320 × 240 resolution from Kinect sensors), these resolutions still allow for the identification of certain characteristics of people (e.g., height, texture, and gender) and, therefore, present privacy problems.Finally, there is no standardized format for presenting fall data. This makes it difficult to use different datasets for application development.

This article presents a new public dataset, which is innovative in at least three aspects. First, it collects data from two different privacy-friendly infrared thermal sensors, with a very low resolution. The low-cost sensors used for this purpose are an Omron D6T-8L-06 and a Melexis MLX90640. Both sensors can be purchased commercially at an approximate value of $52 and $49 dollars, respectively. As far as we know, these sensors have only been used for the detection of falls in older adults [11], but other investigations based on similar sensors have also been carried out [12,13]. Second, it is constructed by two types of volunteers: normal young people (as usual) and performing artists, with the latter group assisted by a physiotherapist to emulate the real fall conditions of older adults. Finally, the types of falls are selected as a result of a thorough literature review. Note that the dataset is limited to the case of one person.

The paper continues as follows. In Section 2, we review 18 public datasets on falls obtained since 2008 from ambient-based sensors. As far as we know, so far, this is the largest survey of datasets based on a vision/camera approach. This gives us an idea of the usual size of the datasets and their main characteristics. In Section 3, we describe the materials (i.e., the two different infrared thermal sensors) and methods used to build the dataset. Here, we also present the details of the new eHomeSeniors dataset. In Section 4, we describe a brief experiment that compares the data obtained by the two thermal sensors and the two types of volunteers. In Section 5, we discuss the main results, and in Section 6, we present the conclusions as well as some ideas of future work.

## 2. Related Work

Although public datasets on falls are still scarce, from 2008 onwards, more and more public datasets have appeared, as well as numerous surveys related to automatic fall detection. Only between 1998 and 2012, a systematic review on automatic FDS using body-worn sensors gathered 96 research papers [14]. In 2015, a survey collected five vision-based public datasets on falls [15], while in 2017, twelve wearable-based public datasets on falls were surveyed [16]. Additionally, only in 2019 have surveys about techniques for abnormal human activity recognition [17], healthcare monitoring systems for elderly people [18], and fall prediction with sensors in smart homes appeared [7].

Using Google Scholar, we collected all the citations of the 2015 survey [15] found until June 2019, as well as all citations to the corresponding datasets included in that survey. The search results were filtered with the keywords “public dataset” + “fall”. From the results obtained, we selected only those publications that published new public datasets on falls obtained from ambient-based sensors. This search process was repeated for each new article found in this way, using a snowballing literature review approach. Thus, we found a total of 18 public datasets on falls based on ambient sensors, published between 2008 and 2019. In addition, it was found the YouTube Fall Dataset (YTFD), created in 2016, but until June 2019, it has not been already published online [19]. As far as we know, this is the largest collection that exists to date on this type of datasets. The results of this search are described in Table 2. Details of the falls collected, of the participants involved in the sample, and of the data collection system used for each case are included.

It is observed that all available fall data have been made by young adults in good health, falling according to their physiognomy and without emulating falls of an older adult. There are other fall datasets that are simply not intended for the fall detection of older adults. For example, in Reference [36], the authors use accelerometers to collect data of falls simulated by practitioners of the athletic discipline parkour. In general, it is also observed that most investigations construct datasets to be used as benchmarks in FDS based on traditional supervised techniques (e.g., threshold based and machine learning). Therefore, together with fall actions, several of these datasets also include data of activities of daily living, useful for training of their algorithms. These additional activities usually involve actions such as walking, sitting, standing, etc., and they do not imply additional technical or ethical difficulties. In fact, they are very simple data to generate and emulate automatically. That is why, in this article, we focus exclusively on the actions of falls.

Note that, on average, datasets include 121 falls of 4 types (they may be different from each other) made by 12 volunteers. The median is 60 falls, 4 fall types, and 10 volunteers. Among the ambient-based devices used, the most common are Kinect sensors (9 datasets), followed by cameras (6 datasets), and infrared thermal sensors (4 datasets).

In Table 3, we include the 30 types of falls used by these datasets for those cases in which more than one type of fall is specified. The classification of the first column was created for this work in order to better organize the different types of falls. In the same table, we also include 14 additional fall types chosen for the SisFall dataset [9]. This is a well-known dataset based on wearable devices. We include it here because it uses 15 types of falls, chosen from 41 types of falls used by another study [37], crossed with a survey of 15 seniors living alone and 17 administrative people from retirement homes. This article is the only one from the table that considers more detailed falls caused by fainting (syncope or falling asleep). On average, each dataset uses 6 types of falls, with a median of 5. The most commonly used types of falls are backward (from standing), lateral (from standing), backward when trying to sit down (empty chair), and forward (from standing).

## 3. Materials and Methods

In this section, the process carried out to build the eHomeSeniors dataset is described in detail. Section 3.1 describes the two different infrared thermal sensors used to collect data. Section 3.2 describes the methodological process for data collection, including the selection of sample size, number of fall types, and volunteers. Finally, Section 3.3 describes the dataset in detail, including its information for download and operation.

### 3.1. Data Collection Systems

The first sensor used in the dataset is a Melexis MLX90640 Far Infrared Thermal Sensor. It is a low-cost sensor that contains 768 FIR (Far Infrared) pixels and provides a privacy-friendly, low-resolution image of 32 × 24 pixels, with a frame rate of approx. 16 fps. It has an operational temperature range between −40 °C and 85 °C and can measure object temperatures between −40 °C and 300 °C. Figure 1 shows two example frames collected by this sensor, painted according to the temperature range of each pixel. Note that the image quality clearly distinguishes a fall, and at the same time, it fully conserves the privacy of the person. This sensor was fixed to a wall at a height of 1.2 m, so that the viewing angle is distributed equally from the center of the sensor, forming a vertical angle of 55° and a horizontal angle of 37.5° as shown in Figure 2.

The second sensor is simpler than the previous one, since the returned heat maps are distributed in linear arrays instead of two-dimensional arrays. It is an Omron D6T-8L-06 infrared thermal sensor. It provides a very low-resolution image of just 1 × 8 pixels. It has an operational temperature range between 0 °C and 50 °C and can measure object temperatures between −10 °C and 60 °C. In order to identify falls and to expand the opening range, we use a system of four of these sensors: two sensors at half height (1 m from the floor) and two sensors at ground level (10 cm from the floor). The four sensors are connected to an ATMEGA328P microcontroller, which reads sensor data and sends it to an ODROID-C1+ via UART interface with a baud rate of 115, 200 and a sample rate of 5 Hz. In this way, a fall is recognized as a decrease in the temperature identified by the upper sensors and an increase in the temperature of the lower sensors. The dataset collects data from this four-sensor system, with a frame rate of approx. 5 fps. This sensor system was previously used in preliminary laboratory experiments, obtaining 93% accuracy in fall detection for a neural network based on a bi-LSTM model (bidirectional long-term memory) [11]. Figure 3 illustrates the temperatures detected by this sensor system for a standing body (left) and a fallen body (right). Note that, when the body reaches the ground, the temperature is concentrated in the lower sensors. In this case, the image quality also conserves the privacy of the person. However, to interpret a fall here is necessary to analyze timestamps.

A schema of the data-collection environment with the two types of sensors is illustrated in Figure 2. Figure 4 shows the real environment where the experiments were performed.

### 3.2. Methodology Description

A fall is often seen as an abnormal movement of the activities of daily living (ADL) [38]. Therefore, to train FDS algorithms, datasets usually include both falls and ADLs. In this article, we have focused on the collection of falls, since the ADLs do not require a greater effort and are easily replicable. As a matter of fact, the data collected at the time before each fall can be considered a common ADL, such as standing, walking, sitting, or lying down. In addition, for any dataset based on video images, it is possible to increase the data using a combination of translation, repetition, and rotation effects. Since a fall can occur anywhere in the room, volunteers simulated falls at different distances between 1 and 5 m from the sensors. The room where the experiment was developed has 6 m × 5 m of space and had no furniture inside, except for a chair and a settee bed in the center used in some tests for simulating falls.

In order to choose the types of falls to be included in the dataset, Table 3 and expert knowledge of a physiotherapist with experience in elderly care were taken as the starting point. After a first review of the list, 25 types of falls were discarded (57% of the list): F1–F8, F20, F21, F33–F36, F38, and F43 for being too general; F13 for being very unlikely in the context of an older adult in a closed space; F24–F26 for being more unnatural at the kinesthetic level; F27 and F28 because they are much less risky than falls on the ground; and F30–F32 because the severity of the impact is irrelevant for the purpose of detecting falls (we must recognize them all, regardless of their severity). However, from these discarded fall types, three new types emerge, not considered in the table: “Backward (from walking backward)” (from F5), “Falling from bed” (from F27 and F28), and “Backward (from standing; knee flexion; slow)” (from F43). From the three, the first two have been considered in other studies, related to wearable devices [37,39], and the third one is a combination of “Backward (from standing; slow)” [40] and “Backward (knee flexion)” [37].

To choose among the remaining 19 types of falls, a small pilot experiment was performed under laboratory conditions with the two infrared thermal sensors to observe the images generated by each type of fall. In this way, considering the image similarity between some types of falls, some of them were merged: F9 and F12; F22 and F23; F10 and F39; F11, F41, and F42; and F14, F15, F18, and F19. Finally, of the six remaining types of falls, F40 remained unchanged; F16, F17, F29, and F44 were adjusted; and F37 was divided into two (at normal speed and at slow speed, the latter being more typical in older adults [40]). Note that, although falls with stretched legs may seem forced for young and healthy people, they are very common in older adults with mobility problems. In conclusion, for the new dataset, the following 15 types of falls are considered:Backward (from walking backward)Forward while walking caused by a tripCause by fainting (slow lateral)Backward when trying to sit down (empty chair)From bedBackward (legs straight)Forward (legs straight)Forward (knee flexion)Backward (from standing; knee flexion, slow)Forward (from standing; knee flexion; slow)Lateral (from standing; legs straight)Lateral (from standing; knee flexion, slow)Cause by fainting or falling asleep (slow backward)Cause by fainting or falling asleep (slow forward)From chair, caused by fainting or falling asleep

Note that the number of fall types considered for this dataset equals the maximum number of those considered in Table 3 and exceeds diversity of all datasets based on ambient-based sensors summarized in Table 2.

Regarding the volunteers for the falls, we used two groups of three people each (see Table 4). Group 2 is made up of young and healthy people, as usual. Group 1, instead, is made up of three performing artists who work on a contemporary dance piece related to the concept of “falling”. Group 2 did not receive any type of instruction. Only each type of fall they had to make were mentioned. Group 1 was assisted by the physiotherapist to instruct them about the differences in the way of falling for an older adult with respect to a young and healthy person.

Since falls occur due to a mismatch between an individual’s physiological function, environmental requirements, and the individual’s behaviour [41], we considered the differences of an older person in two of these three aspects to perform a likely elder’s fall. In the physiological function, all senses are involved in maintaining an active attention required to prevent a fall, so sensory decline may result in impaired perception of environmental challenges [41,42]. The proprioception or awareness of where body parts are in space and the reaction time to respond to unexpected perturbations may be altered, changing the capacity to react. Also, muscle strength may be diminished or altered due to inactivity, which may lower the ability to extend the legs against gravity, making regaining an upright position in the case of a trip more difficult. Function of the various components of successful postural control can be adversely affected by physiological aging and low levels of appropriate physical activity due to disuse. Patients with osteopenia may have bone fractures or injuries as a result of low-energy trauma, typically a fall from standing height or less [43]. Furthermore, there is a disproportionately higher number of deaths in elderly compared to young people as a consequence of a same-level fall [44].

Even with standard bone density, the most common serious injury associated with the fall of an elderly person is a hip fracture, which is associated with up to 20% chance of death and 25% chance of long-term institutionalization [45]. Acute medical problems like infections, chronic conditions such as diabetes, and progressive conditions such as Parkinson’s disease can also affect postural control/balance. There is an impact of medications on successful postural control, with psychoactive medications being particularly associated with falls. Another important aspect is cardiovascular and respiratory correct function, which ensures oxygen transport to the muscles and the brain to enable these functions to occur, and these can be also altered in elders due to disease or as an effect of aging. About environmental requirements, an older person with impaired physiology may fall in an unchallenging environment, which is considered in the way the fall was performed. Since individual’s behaviours are specially subjective, we considered it indirect to the performance of falls in a laboratory environment and not measurable for this particular case.

Each volunteer made 5 falls of each type. From all of them, only the last two falls (volunteer 6, group 2) taken with the Omron sensors presented problems and had to be discarded. This is enough to obtain a total of 15×5×6−2=448 falls in total, which makes it a larger fall dataset than all of those summarized in Table 2. Recall that the average between all the datasets summarized in Table 2 is 121 falls, with a median of 60.

### 3.3. eHomeSeniors Dataset Description

The dataset is publicly available (see the files in Appendix A). It is made up of 180 files in .csv format, one for each fall type. The name of each file follows the form “sensor_name-G*X*-*Y*-fZZ”, where sensor_name is either omron or melexis; *X* is either 1 or 2, so that G*X* represents the number of the group; *Y* is a number between 1 and 6, the number of the volunteer; and ZZ is a number between 1 and 15, so that fZZ represents the number of the fall type. Each file contains five falls, except omron-G2-3-f15, that only contains three.

The Omron sensor files are simpler. Each row contains 33 values separated by semicolons. The first value contains the date and time of collection of the data. Each of the following 16 values includes a decimal value, which represents the temperature collected by the pixels of the two upper Omron sensors, and the last 16 values contain the temperature collected by the lower sensors. Thus, each row represents a frame, which when visualized as a heat map constitutes a different moment of a fall.

The Melexis sensor files are a bit more complex. For each row, the first value contains the data collection time and the second value is information about the sensor model. The following 768 values contain the temperature of each of the pixels that make up a 32×24 pixel heat image. Finally, each row contains several additional data with the raw data from which the temperatures are obtained through formulas documented for the sensor.

Since the Melexis sensor files contain raw data in addition to temperatures, have more pixels, and also have more fps than the Omron sensor, the files are much heavier. The files of the Omron sensors total 8.18 MB, while that of the Melexis totals 802 MB. In addition to these files, the dataset also contains the same data in Matlab numeric matrix .mat format.

## 4. Experimental Results: Estimation of the Fall Duration

In order to investigate the differences between group 1 (artists mimicking elderly falls) and group 2 (young and healthy people), a preliminary heuristic approach has been tested. This approach focuses on the Melexis sensor and is divided in different steps. First of all, pixel positions associated with the volunteer were obtained by considering only those with temperatures above a threshold higher than the background average. The aim of this section is to propose a rather simple approach to verify if there is a statistically significant difference in the fall duration between the two groups.

Figure 5 illustrates the retained pixels of a volunteer in three frames during the realization of a fall. In this case, the values of the threshold and the background were equal to 21 °C and 18.9±0.5 respectively. Then, the barycenter coordinates [x(t),y(t)] of the retained pixels, corresponding to the median position in both direction, was calculated for each time frame. Note that the median was preferred to the average in order to reduce the influence of possible isolated caloric pixels not eliminated in the previous step. It can be observed how the barycenter, indicated with a circle on Figure 5, moved to the right and decreased along the vertical axis. The next step consisted in the smoothing of the barycenter temporal trajectory using a 10-time-step averaging moving window. The final step corresponded to the actual fall detection associated to negative time derivation of the filtered vertical position, i.e., dy/dt<0. Note that falls smaller than 2 pixels were removed. Finally, the fall duration corresponds to the number of successive temporal frames, associated with negative derivation, multiplied by the sampling times, i.e., 1/16 s.

Figure 6 shows the temporal variations of the barycenter positions of five successive type 8 (forward, knee flexion) falls made by volunteers 2 (from group 1) and 5 (from group 2). The upper frame represents the variation of the horizontal barycenter position x(t) through time, while the middle frame represents the variation of the vertical barycenter position y(t). The lower frame represents the spatial trajectories of each falls, the vertical position in function of the horizontal one.

Finally, the histograms of Figure 7 were obtained with the fall durations for all falls of each group. It can be observed that the falls of group 1 (mean 2.62 s) are longer than those of group 2 (mean 2.20 s), in agreement with the fact that volunteers of group 1 were mimicking elderly falls. The two normal distributions were found statistically different $\left(p<10^{-8}\right)$ using a two-sample *t*-test.

## 5. Discussion

Now we shall discuss the results obtained in Section 4. As we mentioned, the first group of volunteers was formed by three performing artists imitating the movements of older adults, while the second group of volunteers were healthy, young people, as is usual in all fall datasets. Through an analysis of the data, it was possible to show that the volunteers of group 1 fell on average more slowly than the volunteers of group 2, which corresponds to the acceleration differences between a healthy, young person and an older adult. It is important to clarify that the presented dataset involves only one person in the sensor’s range of vision, since it is intended to be used by algorithms that interact with people living alone like many elderly people today.

It is worth mentioning that this type of sensor generate a lot of noise if there are other sources of heat in the radius of vision, such as stoves or pets. In the same way, the temperature of an individual cannot be recognized if his/her body is hindered by another object. Furthermore, it is necessary to mention that infrared sensors are very sensitive to several factors, such as the person’s temperature, his/her clothing, his/her position with respect to the sensors, etc.

## 6. Conclusions

In this article, we have built a public dataset on falls obtained by two different types of thermal sensors. This dataset is novel in several ways. First, unlike many other datasets, the low resolution of these sensors prevents distinguishing physiological characteristics of individuals, which favors their privacy. Secondly, the selected falls were obtained from an exhaustive state of the art added to expert knowledge by a physiotherapist with experience in working with older adults. Third, half of the volunteers chosen for the data collection of falls are performing artists with experience in body work and who were told how to represent the falls of an older adult. As far as we know, this is the first public dataset on falls built by performing artists emulating falls of older people.

As future work, additional comparative analyses between both groups could be performed. It would also be interesting to include obstacles and other heat sources in the images to see how this affects the calculation of the barycenters and the trajectories of the falls. In addition, with the advancement of technology, sensors with improved sensitivities and very small sizes can be used in the detection of falls or ADL, such as barometers of which sensitivity has reached the magnitude of millimeters [46]. Having a dataset with this new generation of sensors can help improve systems and algorithms to help older adults.

## Figures and Tables

**Figure 1 sensors-19-04565-f001:**
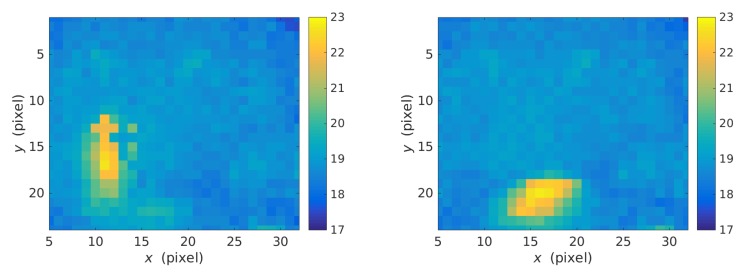
Heat maps of two 32 × 24 frames generated by the Melexis MLX90640 sensor for a standing body (left) and a fallen body (right).

**Figure 2 sensors-19-04565-f002:**
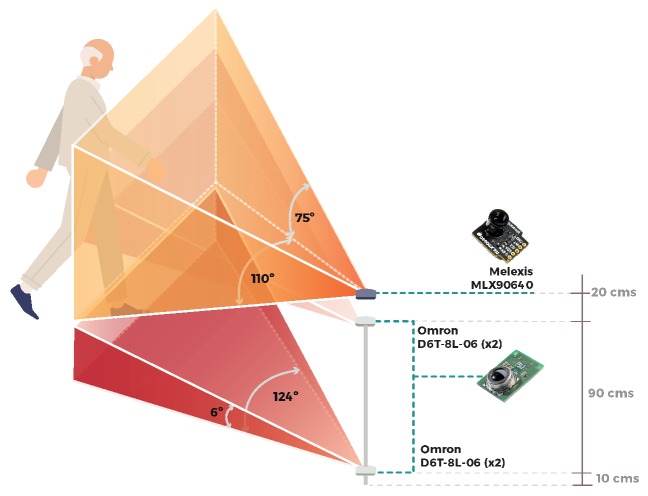
A schema of the data-collection environment with the two types of sensors used for the eHomeSeniors dataset.

**Figure 3 sensors-19-04565-f003:**
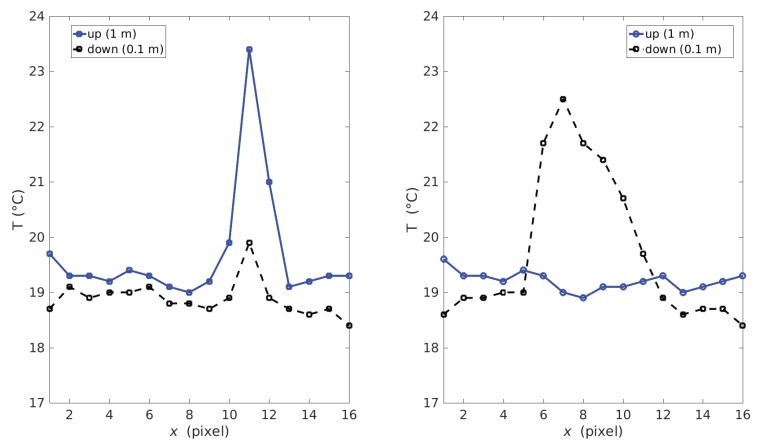
Temperatures detected by the Omron D6T-8L-06 sensor system for a standing body (left) and a fallen body (right): The continuous line corresponds to the 16 pixels placed at 1 meter from the floor and the dashed one corresponds to that at 0.1 m from the floor.

**Figure 4 sensors-19-04565-f004:**
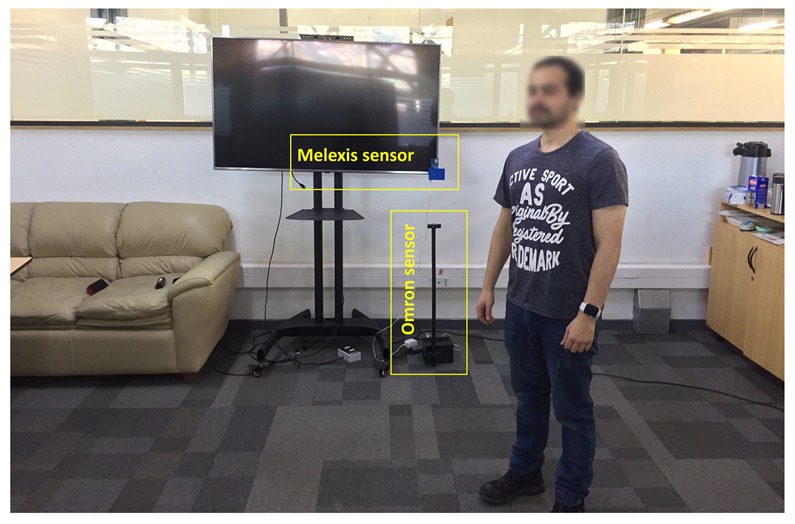
Laboratory where the experiments were performed.

**Figure 5 sensors-19-04565-f005:**
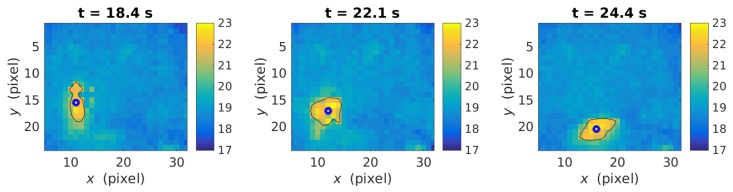
Three different moments during a fall. The blue circle is the barycenter.

**Figure 6 sensors-19-04565-f006:**
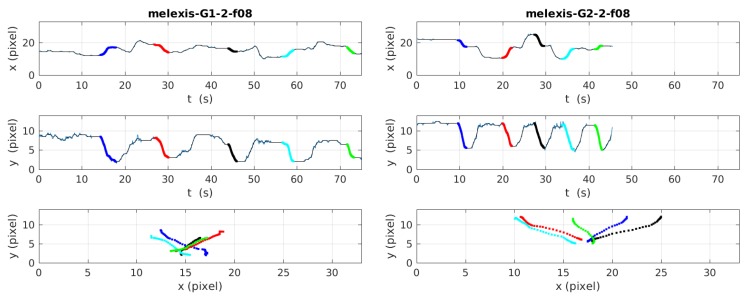
Trajectories of two different volunteers during the falls of type 8 (forward, knee flexion).

**Figure 7 sensors-19-04565-f007:**
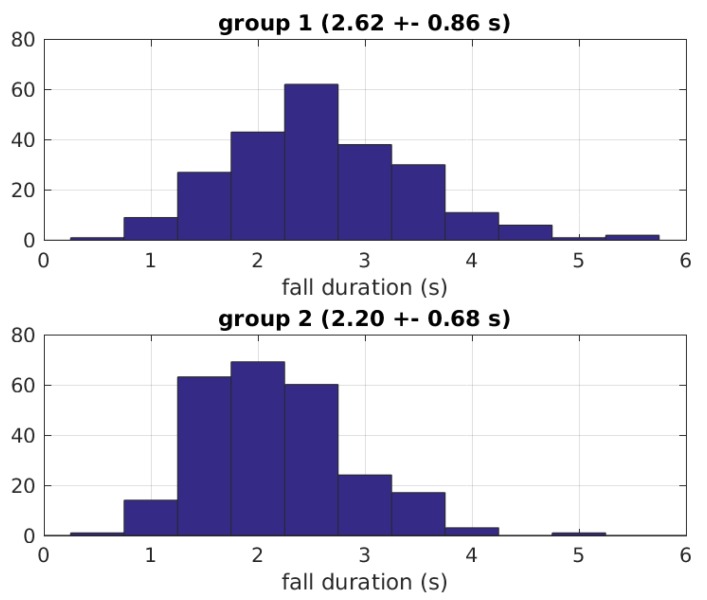
Seconds per fall for each group.

**Table 1 sensors-19-04565-t001:** The main types of devices that collect fall datasets and their positive and negative aspects.

Devices	Examples	Type of Data	Positive	Negative
Wereable devices	smartwatch, smartphone (compass, accelerometer, magnetometer, and gyroscope), inertial measurement unit (IMU), and EEG	acceleration, orientation data, rotation data, angular velocity, magnetic signal, and brain electrical activity	privacy-friendly, rich data, and highly accurate performance	invasive and depends on both the user’s memory and abilities to use them all the time.
Ambient-based sensors	camera, Kinect sensor, infrared thermal sensor, and pressure sensor (on the floor),	low-resolution video, low-resolution image (RGB, depth, or skeleton data), and ambient light	noninvasive, user independence, and long battery life	intrusive (it depends on resolution and data quality); only suitable for closed spaces; noise from other objects, people, or pets.

**Table 2 sensors-19-04565-t002:** Public datasets on falls obtained from ambient-based sensors.

Year	Dataset Name	Ref.	Falls		Participants				Data Collection Systems
			#	#types	#	#F	#M	Age	
2019	UP-Fall	[20]	255	5	17	8	9	18–24	6 infrared sensors, 2 cameras (18 fps), 5 IMUs with accelerometer, gyroscope, ambient light, 1 EEG
2018	CMDFALL	[21]	400	8	50	20	30	21–40	7 overlapped Kinect sensors and 2 WAX3 wireless accelerometers
	FALL-UP	[22]	255	5	17	?	?	?	6 infrared sensors; 2 cameras; 1 EEG; 5 wearable inertial sensors on left ankle, right wrist, neck, waist, and right pocket with accelerometer, angular velocity, and luminosity
	UP-Fall	[23]	60	5	4	2	2	22–58	4 ambient infrared presence/absence sensors, 1 RaspberryPI3, 4 IMUs with accelerometer, ambient light, angular velocity, 1 EEG
2017	Dataset-D	[24]	95	2	4	?	?	30–40	4 Kinect sensors (RGB, depth, skeleton data; 20 fps, 640 × 480)
	MICAFALL-1	[24]	40	2	20	?	?	25–35	*idem*
	Thermal Simulated Fall	[8]	35	?	?	?	?	?	9 FLIR ONE thermal cameras (640 × 480) mounted to Android phone
2016	KUL Simulated Fall	[25]	55	?	10	?	?	?	5 web-cameras (12 fps, 640 × 480)
2015	–	[26]	21	4	?	?	?	?	IP camera (Dlink DCS-920) through Wi-Fi connection (MJPEG, 320 × 240)
	EDF	[15]	320	?	10	?	?	?	2 Kinect sensors (depth maps, 320 × 240, 30 fps)
2014	OCCU	[27]	60	1	5	?	?	?	*idem*
	SDU Fall	[28]	30	1	10	2–8	2–8	young	1 Kinect sensor
	TST	[29]	132	4	11	?	?	22–39	1 Kinect sensor (depth maps); 2 IMUs on waist and right wrist with accelerometer
	UR Fall	[30]	30	2	5	0	5	>26	2 Kinect sensors (depth maps); 1 IMU on waist (near the pelvis) with accelerometer
2013	Le2i fall	[31]	143	3	11	?	?	?	1 video camera in 4 different locations (25 fps, 320 × 240)
2012	Le2i fall	[32]	192	3	11	?	?	?	*idem*
	vlm1	[33]	26	?	6	?	?	?	2 Kinect sensors (RGB, depth; 10 fps, 320 × 240)
2008	Multi camera fall	[34,35]	22	8	1	0	1	adult	8 video cameras
average			121	4	12				
median			60	4	10				

**Table 3 sensors-19-04565-t003:** Classification of different types of falls considered in the literature.

Fall			Reference											
by	ID	Description	[34]	[32]	[29]	[30]	[26]	[9]	[24]	[23]	[22]	[21]	[20]	#
general	F1	Fall (from standing)	✗	✗	✗	✓	✗	✗	✓	✗	✗	✗	✗	2
	F2	Backward (from standing)	✗	✗	✗	✓	✓	✗	✗	✓	✓	✗	✓	5
	F3	Forward (from standing)	✗	✓	✗	✓	✓	✗	✗	✗	✓	✗	✗	4
	F4	Lateral (from standing)	✗	✗	✗	✓	✓	✗	✗	✓	✓	✗	✓	5
	F5	Backward (from walking)	✗	✗	✗	✗	✗	✗	✗	✗	✗	✓	✗	1
	F6	Forward (from walking)	✗	✗	✗	✗	✗	✗	✗	✗	✗	✓	✗	1
	F7	Leftward (from walking)	✗	✗	✗	✗	✗	✗	✗	✗	✗	✓	✗	1
	F8	Rightward (from walking)	✗	✗	✗	✗	✗	✗	✗	✗	✗	✓	✗	1
cause	F9	Forward while walking caused by a slip	✗	✗	✗	✗	✗	✓	✗	✗	✗	✗	✗	1
	F10	Backward while walking caused by a slip	✗	✗	✗	✗	✗	✓	✗	✗	✗	✗	✗	1
	F11	Lateral while walking caused by a slip	✗	✗	✗	✗	✗	✓	✗	✗	✗	✗	✗	1
	F12	Forward while walking caused by a trip	✗	✗	✗	✗	✗	✓	✗	✗	✗	✗	✗	1
	F13	Forward while jogging caused by a trip	✗	✗	✗	✗	✗	✓	✗	✗	✗	✗	✗	1
	F14	Cause by fainting/syncope/loss of balance	✗	✓	✗	✗	✓	✗	✗	✗	✗	✗	✗	2
	F15	Vertical fall while walking, by fainting	✗	✗	✗	✗	✗	✓	✗	✗	✗	✗	✗	1
	F16	Forward while sitting, caused by fainting	✗	✗	✗	✗	✗	✓	✗	✗	✗	✗	✗	1
	F17	Backward while sitting, caused by fainting	✗	✗	✗	✗	✗	✓	✗	✗	✗	✗	✗	1
	F18	Lateral while sitting, caused by fainting	✗	✗	✗	✗	✗	✓	✗	✗	✗	✗	✗	1
	F19	Fall while walking caused by fainting	✗	✗	✗	✗	✗	✓	✗	✗	✗	✗	✗	1
		(use of hands in a table to dampen fall)
	F20	Forward when trying to get up	✗	✗	✗	✗	✗	✓	✗	✗	✗	✗	✗	1
	F21	Lateral when trying to get up	✗	✗	✗	✗	✗	✓	✗	✗	✗	✗	✗	1
	F22	Forward when trying to sit down	✗	✗	✗	✗	✗	✓	✗	✗	✗	✗	✗	1
	F23	Backward when trying to sit down	✓	✓	✗	✗	✗	✓	✗	✓	✗	✗	✓	5
	F24	Lateral when trying to sit down	✗	✗	✗	✗	✗	✓	✗	✗	✗	✗	✗	1
	F25	Leftward when trying to sit down	✗	✗	✗	✗	✗	✗	✗	✗	✗	✓	✗	1
	F26	Rightward when trying to sit down	✗	✗	✗	✗	✗	✗	✗	✗	✗	✓	✗	1
location	F27	On bed (then leftward)	✗	✗	✗	✗	✗	✗	✗	✗	✗	✓	✗	1
	F28	On bed (then rightward)	✗	✗	✗	✗	✗	✗	✗	✗	✗	✓	✗	1
	F29	From chair	✗	✗	✗	✓	✗	✗	✓	✗	✗	✗	✗	2
impact	F30	Fall (impact on hands and elbows)	✗	✗	✗	✗	✗	✗	✗	✓	✗	✗	✗	1
	F31	Forward (impact on hands and elbows)	✗	✗	✗	✗	✗	✗	✗	✗	✗	✗	✓	1
	F32	Forward (impact on knee)	✗	✗	✗	✗	✗	✗	✗	✓	✗	✗	✓	2
termination	F33	Backward (end up sitting)	✗	✗	✓	✗	✗	✗	✗	✗	✗	✗	✗	1
	F34	Backward (end up lying)	✗	✗	✓	✗	✗	✗	✗	✗	✗	✗	✗	1
	F35	Forward (end up lying)	✗	✗	✓	✗	✗	✗	✗	✗	✗	✗	✗	1
	F36	Lateral (end up lying)	✗	✗	✓	✗	✗	✗	✗	✗	✗	✗	✗	1
	F37	Forward on knees (stay down)	✗	✗	✗	✗	✗	✗	✗	✗	✓	✗	✗	1
articulation	F38	Fall (legs straight)	✓	✗	✗	✗	✗	✗	✗	✗	✗	✗	✗	1
	F39	Fall Backward (legs straight)	✓	✗	✗	✗	✗	✗	✗	✗	✗	✗	✗	1
	F40	Fall Forward (legs straight)	✓	✗	✗	✗	✗	✗	✗	✗	✗	✗	✗	1
	F41	Fall Leftward (legs straight)	✓	✗	✗	✗	✗	✗	✗	✗	✗	✗	✗	1
	F42	Fall Rightward (legs straight)	✓	✗	✗	✗	✗	✗	✗	✗	✗	✗	✗	1
	F43	Fall Fall (knee flexion)	✓	✗	✗	✗	✗	✗	✗	✗	✓	✗	✗	2
	F44	Fall Rightward (knee flexion)	✓	✗	✗	✗	✗	✗	✗	✗	✗	✗	✗	1
# fall types			8	3	4	5	4	15	2	5	5	8	5

**Table 4 sensors-19-04565-t004:** General volunteer features for the falls collection: Group 1 is formed by performing artists and group 2 is formed by normal, healthy, young people.

	#	Gender	Age	Weight	Height
group 1	1	F	37	59	1.64
	2	F	34	51	1.62
	3	M	35	62	1.80
group 2	4	F	27	49	1.52
	5	M	28	89	1.73
	6	M	29	66	1.65

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
