# Peer review of "eHomeSeniors Dataset: An Infrared Thermal Sensor Dataset for Automatic Fall Detection Research"

_sensors, 2019, doi:10.3390/s19204565_

Round 1

Reviewer 1 Report

Summary:

- The paper presents the process of constructing a falls dataset with data obtained from infrared sensors. Data from this dataset can be used by proposed fall detection systems experiments. The main differentials of this dataset for other public datasets are: (i) it collects data from different privacy-friendly infrared thermal sensors; (ii) it is constructed by normal young people (as usual) and performing artists, being the latter group assisted by a physiotherapist to emulate the real fall conditions of older adults; (iii) the types of falls executed are selected from a literature review.

Pros:

- The paper presents interesting and relevant research, since there is a clear demand for data for fall detection solution experiments.

- The Dataset is well described and the way it was built is clear. The concern with the profile of participants, especially participants artists, who tried to emulate falls from the older adults, shows the commitment to build a robust dataset.

Cons:

- Lack of detail on the literature review performed. One cannot replicate it with the information provided in the study. And since the authors cite that this review was systematic (see Table 3), a better clarification was expected to allow replication.

- Still on the review, Lines 84 and 85 describe “This search process was repeated recursively for each new article found in the way”. So, was a snowballing performed in this process? This was not very clear in the paper.

- There is not much discussion about the impact of weight and height on experiments. Since the number of participants is small, it would be good to clarify if there is a real impact of this profile that could lead to a discrepancy in testing with this dataset in another research. Especially in group 1, where the performing artists are, there is not much variation in the weight of all volunteers, and height of the two female participants is also very close, while the male participant is much bigger. It does not appear to be a regular distribution of the weight and height profile of these volunteers.

Suggested Improvements:

- The article uses two specific types of infrared sensors. There is then a need to make clear if these sensors are commonly used by other research. Are they easily accessible? Is it possible to use other infrared sensors calibrated for the same image quality for experiments based on this dataset? The purpose of these questions is the difficulty of understanding if this dataset is suitable for use by other research, or if there is a difficulty related to these sensors that makes it impossible to use it by others.

- There is also a need to better detail how the categories (general, cause, location, impact, termination, articulation) of the fall types of table 3 were chosen? Is it directly from the systematic review?

- It would be interesting if the movements of the day were not completely discarded (walking, running, jumping). It is stated in the paper that there are datasets of older adults performing these types of movements. These datasets could be compared with the performance of the artists for a possible increase in the reliability of the movements made by them in relation to the elderly.

- The authors should have a look at a study that deals with fall detection using experiments with parkour athletes, which  is: 

Bulotta, S., Mahmoud, H., Masulli, F., Palummeri, E., & Rovetta, S. (2013). Fall detection using an ensemble of learning machines. In Neural Nets and Surroundings "Fall Detection Using an Ensemble of Learning Machines".

This is not a much-cited paper, but it might be interesting to have a look.

- The authors could improve the discussion of the results and the problems related to the dataset, for example, the situation if there are two persons in the environment, how they can discover who fall is?

- Although the authors do not focus on collecting data of ADL, they could discuss how the dataset could improve the research in this area when comparing the data of ADL. The data collected is only from the sensors used? In the case of using another infrared sensor, how this dataset can help? For example, the data of one sensor contains 33 values, and the second contains a matrix of 32x24. How do you deal with this? Moreover, how this difference can impact in the dataset? 

About the paper writing, in the introduction, references are missing in page 2, second paragraph, and in the second item of this page. Instead of "decreasing", the verb should be "decrease" in the second item. "From the results" in page 3, instead of "of the results" (see also page 8, last paragraph). The authors should review the last paragraph of page 3. In page 10, "volunteers" instead of "volunteer" in the last paragraph and "for all falls" instead of "for the all falls". In page 11, use "his/her" instead of "his".

Moreover, the name of the dataset should appear in the beginning. Commas must be added in page 3, first paragraph. About reference [14] in page 3, what is its connection with datasets? 

Author Response

Thank you very much for your comments.

We attach our response considering the corrections for each of the comments.

Reviewer 2 Report

The papers deals with the generation of a new data set for indoor falls. Authors declare that the novelty is threefold: the use of infrared sensors, the population of volunteers (in particular performing artists assisted by physiotherapist to emulate the real fall conditions of older) and the types of falls selected from a detailed systematic review of the literature).

As far as technologies review, barometers are not mentioned; the latest generation of barometric mems technology is interesting due to its sensitivity. They allow the detection of changes in height of just a few millimeters (https://www.cismst.de/en/aktuelles/presse/presseinfo/presse-2017-05/, “A Wearable Fall Detector for Elderly People Based on AHRS and Barometric Sensor "," Fall Detection Using a Head-Worn "," Falling Detection Using a Head-Worn Barometer ”, etc.).

The most significant criticism for this work, in my opinion, is that the creation of a data set have not to be an end in itself. The work does not specify how data could be used by other research activity. The main impression is that the work was useful to the authors but with different purposes than what the paper would like to be. In my opinion, a data set is evaluated both for its size (how many hours or seconds of activity or the number of scenarios and cases or other), and for the diversity of the scenarios, and for its usefulness; in particular, a data set serves scientist to compare and evaluate new algorithms, methodologies, criteria, technologies, processes...

The data produced is not characterized to be further analyzed; it is not even possible to recreate the same experimental environment. For example, it is not described how sensors are positioned, the distance from the subjects, the environmental conditions (figure 3 is not enough): the experiment is not replicable. Furthermore, it must be taken into account that each cell of the sensor integrates the energy emitted by the bodies; for this reason, the distance from the subject influences the quantization of the information. By changing the subject's position (which also occurs during the fall: the subject moves with respect to the sensitive element) the detected data changes. Furthermore, the sensor in question has a well-defined viewing angle that influences the data collected.

Paragraph 4 describes some operations to identify the center of gravity and filtering; a data set should not be modified to allow the user to experiment with the raw data. The observations of paragraph 4 and the described transformations do not agree with the objective of the work: to create a data-set.

The methodology (section 3.2) is interesting (note: the caption in figure 3 is wrong – three sensors but two types) but is disconnected from the objective. In fact, knowing that "since falls two to measure between individual and physiological function, environmental requirements ..." is irrelevant for the construction of the data set (note: it could be a part of the introduction to the problem, but not of the methodology) and is not even used later as a useful tool for establishing algorithms or methods.

The proposed work, certainly long and complex, is – in my opinion - immature to represent a reference point for the scientific community.

It would be interesting to see only the results relating to the method and type of the proposed sensor to detect indoor falls with particular attention to the confusion matrix.

Author Response

(The authors gave the same response as above.)

Round 2

Reviewer 2 Report

Dear Authors, thanks for the revisions.

I still believe that this data set is "difficult" to use due to the lack of data generality and their narrow field of application/usability (note: the sensors used are very sensitive to various factors including the person's temperature, his/her clothing, his/her position with respect to sensors, its/their distance from the person , etc.)
Although my opinion remains the same, it is more than possible that my opinion and perception is incorrect (although our research group has used infrared array sensors for years). The work done deserves to be published; the interest in these data will be evaluated by the scientific community. 

Author Response

Dear reviewer,

Thank you very much for your comments.

We have included an additional sentence in the Discussion, in order to emphasize the typical sensitivity problems of infrared sensors.

Sincerely yours,

Fabián Riquelme
